# The Impact of Digitalization on Macroeconomic Indicators in the New Industrial Age

Alina Ștefania Chenic [1], Adrian Burlacu [2,*], Răzvan Cătălin Dobrea [3], Laurenţiu Tescan [4], Alin Ioan Creţu [1], Mihaela Roberta Stanef-Puica [1], Teodor Narcis Godeanu [5], Alina Magdalena Manole [1], Daniela Virjan [1] and Nicolae Moroianu [1]

1   Department of Economics and Economic Policies, Faculty of Theoretical and Applied Economics, Bucharest University of Economic Studies, Calea Dorobanti 15-17, 010552 Bucharest, Romania
2   Faculty of Railways, Roads and Bridges, Technical University of Civil Engineering, 122–124 Bd. Lacul Tei, 020396 Bucharest, Romania
3   Department of Maanagement, Faculty of Management, Bucharest University of Economic Studies, Caderea Bastiliei Street No. 2-10, 010374 Bucharest, Romania
4   Ministry of Regional Development and Public Administration, Bld. Libertății No. 16, Latura Nord, Sector 5, 050706 Bucharest, Romania
5   Secretary General of the National Gambling Agency Romania, Calea Victoriei 9, 030022 Bucharest, Romania
*   Correspondence: adrian.burlacu@utcb.ro

**Abstract:** Digitalization has become a watchword in all areas of economic and social activity, and its integration has become a necessity for every state worldwide. If a few years ago we were talking about technological innovation and cutting-edge technologies we are now talking about digital technologies, robotics, big data, and artificial intelligence, and future industrial production will develop in symbiosis with modern information and communication technology. The paper aims to show the impact of digitalization on macroeconomic indicators, including labor productivity, value-added, and value of exports of goods and services, and in this regard, we have built an econometric model to see how digitalization and the evolution of macroeconomic indicators work, and how they will influence the degree of growth and economic development. The research results verify the hypotheses we started from, namely that there is a positive and strong correlation between digitalization and productivity, as well as between digitalization and value-added, and that there is a positive, yet weaker correlation between digitization and exports. The impact of digitalization on macroeconomic indicators extends beyond the theoretical and practical implications, as they influence both the decision-making process of the companies' management and national and European economic policies.

**Keywords:** digitalization; productivity; value added; export; artificial intelligence





## 1. Introduction

The new Industrial Era (the fourth revolution-Industry 4.0) will develop in symbiosis with modern information and communication technology, aiming for interconnected digital systems. Although numerous research on the importance of digital technologies has been conducted, there are not many studies that analyze the strategic factors that help us understand how prepared we are for the fourth industrial revolution. The beginning of the fourth industrial revolution is linked to the ever-changing transformations in economic processes. As Papanyan points out in 2015, "*while economic theory considers technology as a contribution that integrates capital and labor to produce economic results, technology today has a more comprehensive role in production than just an integrative factor*" [1]. Several general trends, such as technological trends (process automation, growing digitalization, robotics, big data, artificial intelligence) or demographic trends (increasing life expectancy and aging population, new media and information) have a major impact on the way societies live and work. By acting together, these trends have stronger and, at the same time, less predictable effects due to the complex interaction between them.

## 2. Literature Review

Digital transformation is a process that requires certain skills regarding the use of new innovative digital technologies to increase economic performance and improve the quality of life. At the same time, quantifying the influence of digital technologies on economic development is one of the important concerns of scientific studies and analyses. Between digitization and economic growth, there is a direct relationship, analyzed in the specialized literature and presented in Table 1.

**Table 1.** Independent and dependent variables.

| Independent Variable | |
|---|---|
| $DG_i$ | Integration of digital technology, for the state $i$, between 2014 and 2019 |
| **Dependent Variable** | |
| $PM_i$ | Labor productivity per person employed and hours worked, for the state $i$, between 2014 and 2019 |
| $VAB_i$ | Gross value added per capita, for state $i$, period 2014–2019 |
| $EX_i$ | Exports of goods and services in millions of euros, for the state $i$, between 2014 and 2019 |

Digitalization was identified as the main significant technological trend that changes society as well as our businesses [2,3]. Nowadays, when our companies are constantly under pressure to avail digital technology and let business models adjust to this new reality [4]. However, even though the transition to digital comes with many benefits, it also requires investment and associated costs as shown by Ahmad, M. and Murray, J. [5], keeping in mind the visible progress of digital technologies, according to Bejtkovský, J. [6]. The question is how digitalization is used and measured by scientists and academics.

Therefore, our main purpose is to illustrate the current state of the art and to offer a higher understanding of the term digitalization [7]. It is interesting that in specialized literature there is much talk about digital transformation, but too little about digitalization. One of the first literature reviews about digital transformation was written by Henriette et al. [8], following the investigations such as Gebayew et al. [9], Reis et al. [10]; Vukšicet et al. [11]. It is well known that the term digital transformation was invented by professionals in business and subsequently studied by academics.

On the other hand, we also know that a large knowledge gap is currently found at the government level, which represents only 1% of the research worldwide [10]. In response to changing expectations, governments change their current way of operation to improve the supply of public services, while the public administrators themselves define digital transformation in their daily practices [12]. In this regard, academics such as Mergelet et al. [13], offer empirical definitions of digital transformation, using expert interviews rather than literature reviews.

In terms of digitalization, we could find few literature reviews, with an emphasis on the effects of the organizational aspects of digitalization [14] and another research by Parida et al. [15], which developed a communicative framework that sets the tone for upcoming research through binding digitalization, innovation of the business model and sustainability in the industrial backgrounds.

## 3. Statistical Correlations between Digital Technology Integration and Labor Productivity, Gross Value Added, and Exports—At EU Level

### 3.1. Relationship Digitalization—Productivity

The indicator chosen to measure productivity is the productivity of labor per employee and per working hour, calculated as the actual output per unit of labor, based on the total number of working hours. According to Eurostat [16], this approach has a certain advantage:

> *"Measuring hourly labor productivity provides a better picture of the evolution of productivity in the economy than labor productivity for each employee, as it eliminates differences in full-time or part-time employment across countries and years."*

The degree of digital integration of companies is directly related to labor productivity (Figure 1). In general, the higher the digital integration of companies, the higher the labor productivity. However, the degree of dispersion is high enough to suggest the complexity of this relationship and the need for further research.

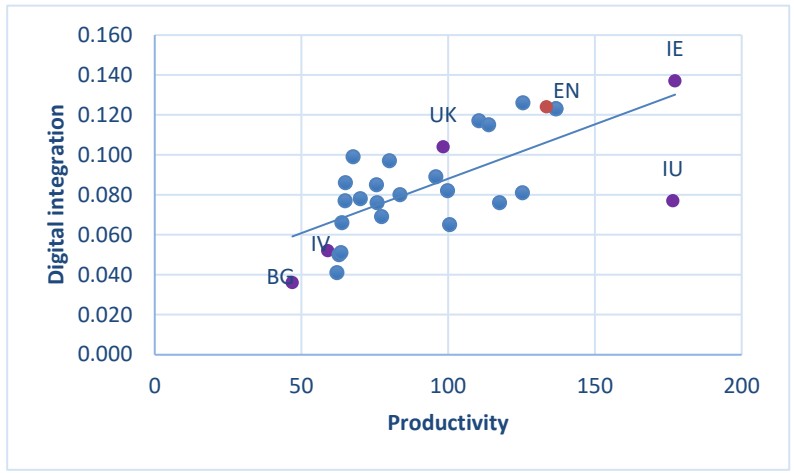

**Figure 1.** The relationship between digital integration and productivity. Source: Processing based on data provided by Eurostat [16].

The higher the percentage of small and medium-sized companies that use a very large number of digital technologies, the higher the level of productivity (Figure 2). At the same time, the higher the percentage of SMEs that use a very small number of digital technologies, the lower the level of productivity. Basically, the positive correlation between digital integration and productivity is maintained when we strictly analyze small and medium-sized companies, with at least 10 employees. The degree of dispersion is slightly higher in the case of very high digital intensity, in relation to productivity. Large companies (Figure 3) indicate a similar relationship, with a clear trend of associating productivity with the intensity of digitalization of companies. In this case, however, a group of three countries (Romania, Bulgaria, and Greece) does not follow the trend in the region, having a very low level of productivity and standing out through the high percentage of large companies with a low degree of digitalization intensity.

Considering the results on productivity in relation to digital integration, on the one hand, and digital intensity in SMEs and large companies, on the other hand, it can be concluded that there is a positive correlation between the digitalization of companies and labor productivity, in which the higher the number of digitalized companies in a state, the higher the labor productivity in that state.

Therefore, the first null hypothesis can be rejected, which means that we can accept the first hypothesis that the higher the level of digitalization, the higher the level of productivity.

All the data sets consulted indicate a low level of productivity and a low degree of digitalization in Romanian companies. Given the positive correlation between the level of digitalization and productivity, investments in the digitalization of Romanian companies can be a solution to the problem of low levels of labor productivity.

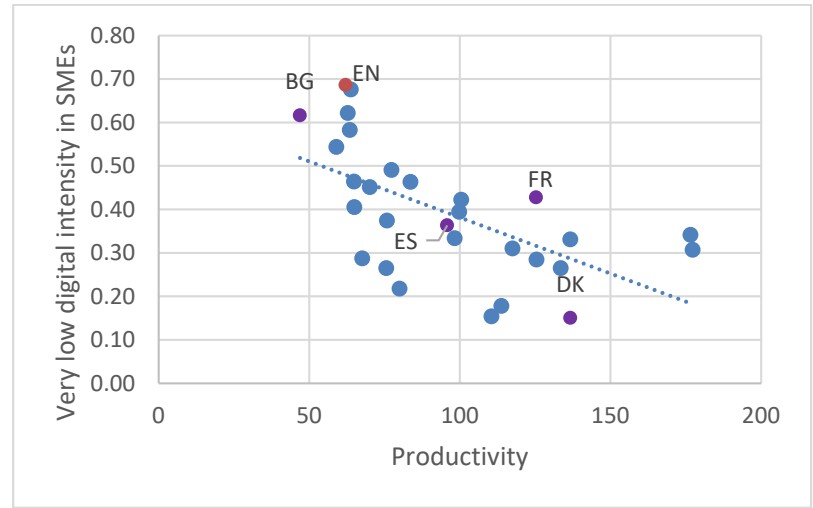

(**a**) Productivity for very low digital intensity in SMEs

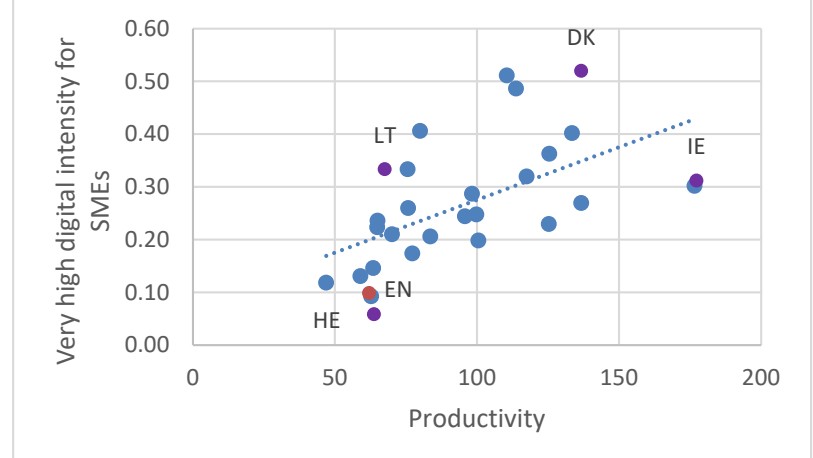

(**b**) Productivity for very high digital intensity in SMEs

**Figure 2.** The relationship between digital intensity in SMEs and the level of productivity. Source: Processing based on data provided by Eurostat [16].

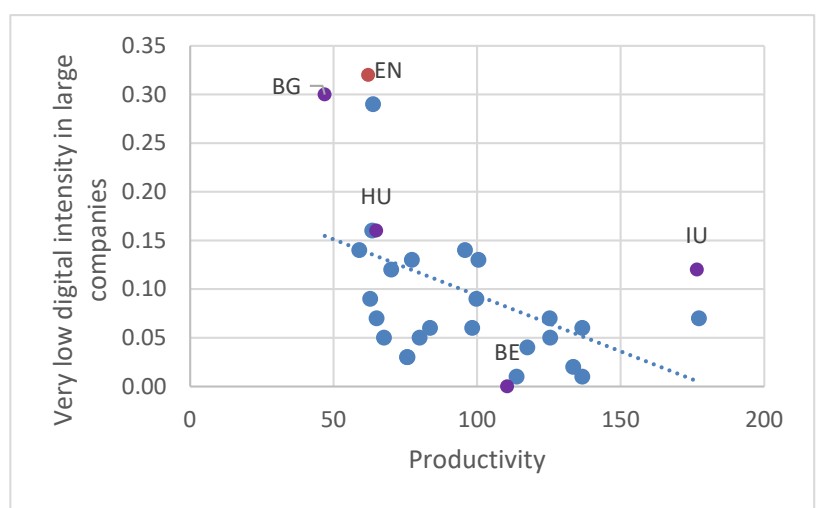

(**a**) Productivity for very low digital intensity in large companies

**Figure 3.** *Cont*.

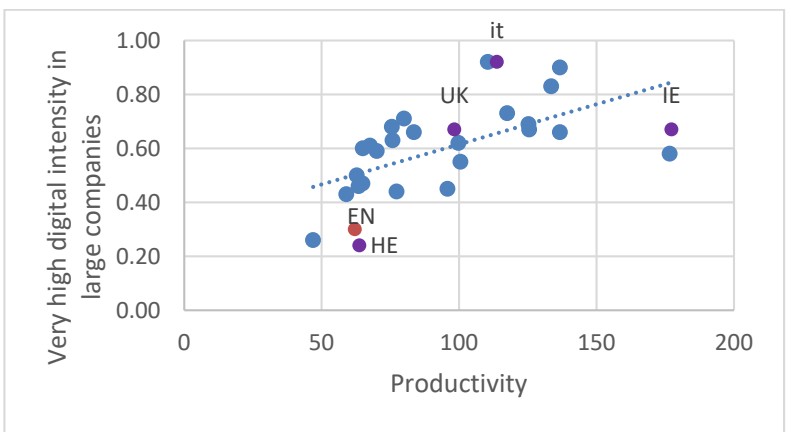

(**b**) Productivity for very high digital intensity in large companies

**Figure 3.** The relationship between digital intensity in large companies and the level of productivity. Source: Processing based on data provided by Eurostat [16].

### 3.2. The Relationship between Digitalization and Value Added

Value-added data were obtained from Eurostat, which provides, on the one hand, Member States' data on gross value added and, on the other hand, data on population, to be able to calculate the value added per capita, so that population differences between EU states should not affect the results obtained. Gross value added (GVA) is expressed in millions of euros and is calculated by the difference between the value of the goods and services produced and intermediate consumption. Figures 4–6 graphically show the relationship between per capita GVA and digitization in the Member States of the European Union [17].

Figure 4, which illustrates the relationship between digital integration and value-added for EU member states in 2019, shows that there is a link between digital integration and value-added, but that this is not very strong, with one exception (Luxembourg, state with high added-value and medium digital integration).

In the case of both SMEs and large companies, a link can be seen between digital intensity and value-added.

The higher the percentage of companies with a very low digitalization intensity, the lower the gross value added per capita. On the contrary, the more consistent the proportion of companies with a very high digitalization intensity, the higher the value added per capita.

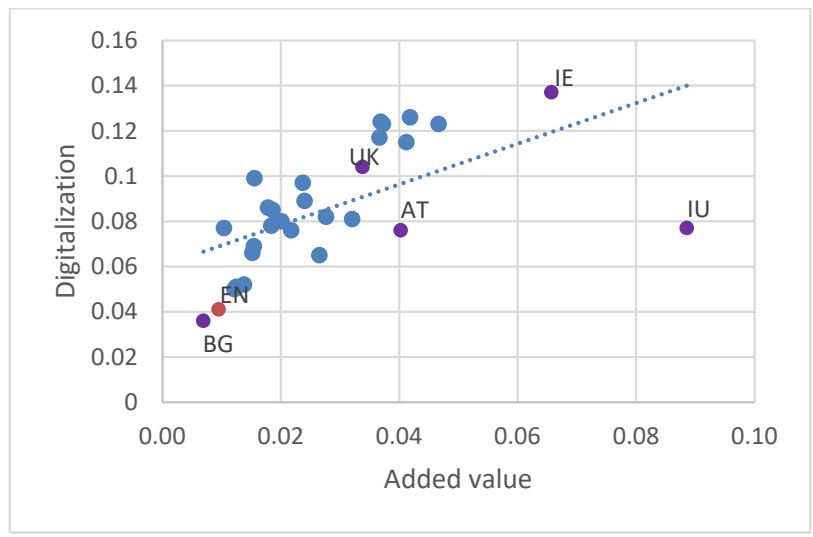

**Figure 4.** The relationship between digital integration and the EU added value in 2019.

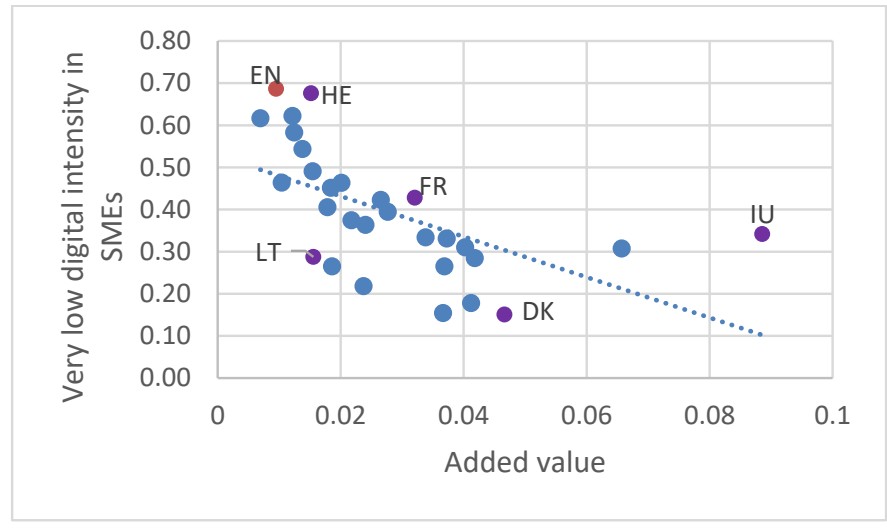

(**a**)     Added value for very low digital intensity in SMEs

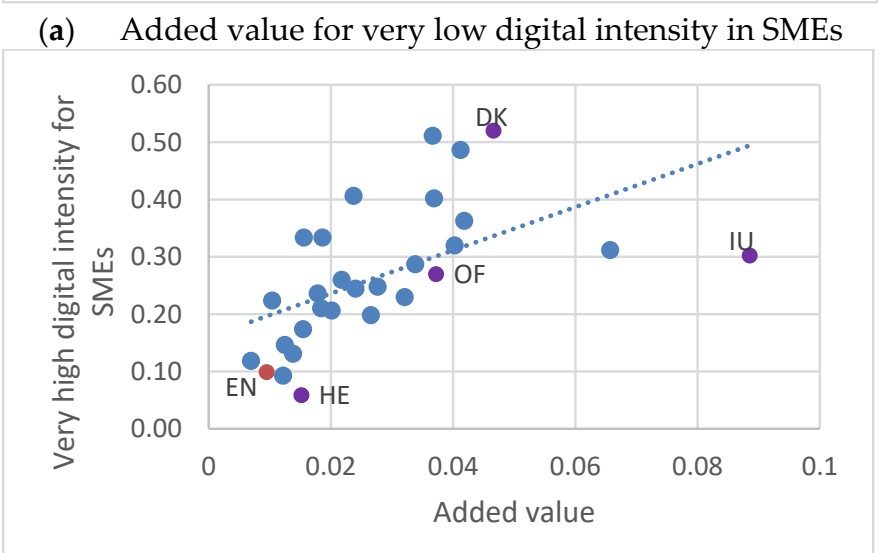

(**b**)     Added value for very high digital intensity in SMEs

**Figure 5.** The relationship between digital intensity in SMEs and EU added value in 2019.

Unlike in the previous case, establishing a causal relationship is not as easy. Considering the studies consulted and the results obtained, we can certainly say that digitalization has a positive effect on added value. However, it is unclear whether the level of value-added does not influence the degree of digitalization of companies—for example, developed countries generally have both higher per capita value-added and better financing mechanisms that allow companies to invest. in the faster implementation of new digital technologies.

Figures 4–6 show a low level of GVA for Romania. Even though the causal relationship between gross value added and digitalization is not clearly established, there is certainly a positive correlation between the two. Additional statistics show that Romania is in second to last place in the EU in terms of the share of SMEs in GVA (32.6%), surpassing only Poland in the ranking (29.6%) [18]. In this context, actions aimed at increasing digitalization for SMEs are even more beneficial and desirable.

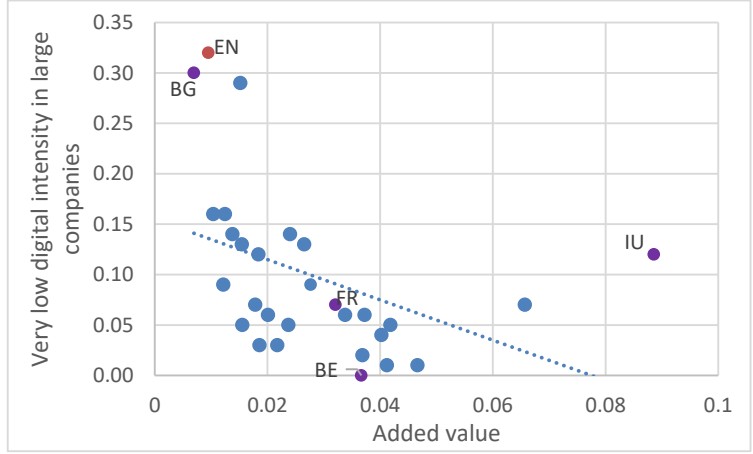

(**a**)     Added value for very low digital intensity in large companies

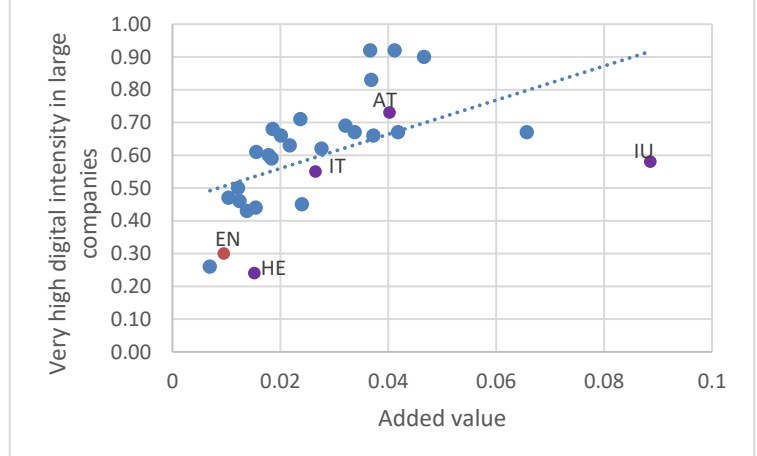

(**b**)     Added value for very high digital intensity in large companies

**Figure 6.** The relationship between digital intensity in large companies and added value in the EU in 2019.

### 3.3. Relationship between Digitalization and Exports

Finally, the relationship between the degree of digitization of companies in the Member States of the European Union and the value of exports for goods and services, expressed in millions of euros, is analyzed (Figures 7–9).

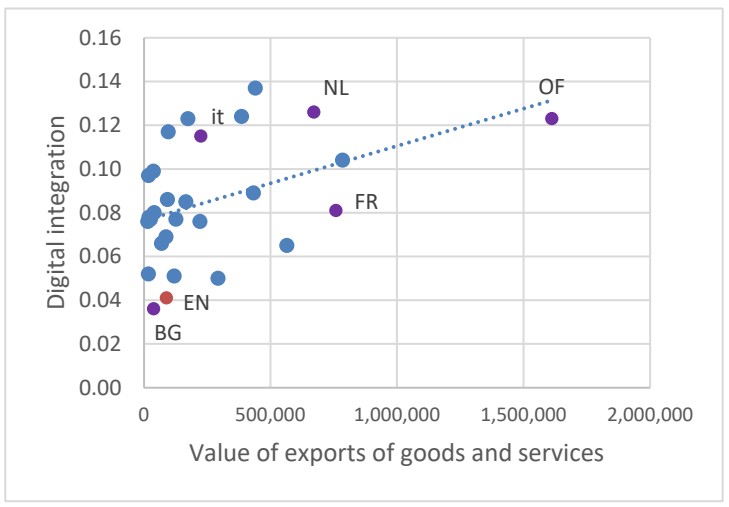

**Figure 7.** The relationship between digital integration and exports was added to the EU in 2019.

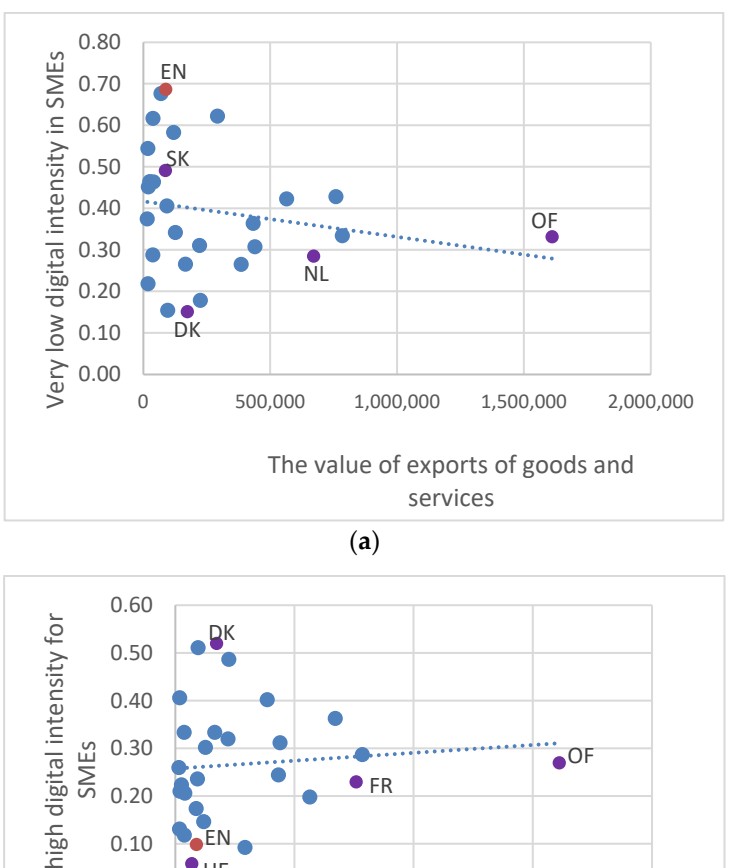

**Figure 8.** The relationship between digital intensity for SMEs and EU exports in 2019. (**a**) The value of exports of goods and services for very low digital intensity in SMEs. (**b**) The value of exports of goods and services for very high digital intensity in SMEs.

When we compare the value of exports with the level of digital integration, we have a high degree of dispersion, but also a clear trend, which suggests a positive correlation between the two variables.

Unlike the first two variables tested, exports have a higher correlation and dispersion in relation to both digital intensity and digital integration. Comparing the results of SMEs with those of large companies, we can see a greater association between the two variables in the case of large companies. The results can be explained by the fact that a very small percentage of companies (especially SMEs) use digital technologies to support the export of products and services. Therefore, their impact on the economy cannot be easily seen. Also, in this case, the results obtained in the case of digital intensity are more robust than in the case of digital integration, because the latter has an indicator of the percentage of companies that use digital technologies for sales abroad, which can affect the results.

The value of exports of goods and services is influenced by several variables, including digitalization, mainly due to e-commerce components. For countries such as Romania, where the value of exports and digitalization are low compared to other European countries, investing in developing the capacity of e-commerce companies is an opportunity to effectively increase exports, both in volume and value.

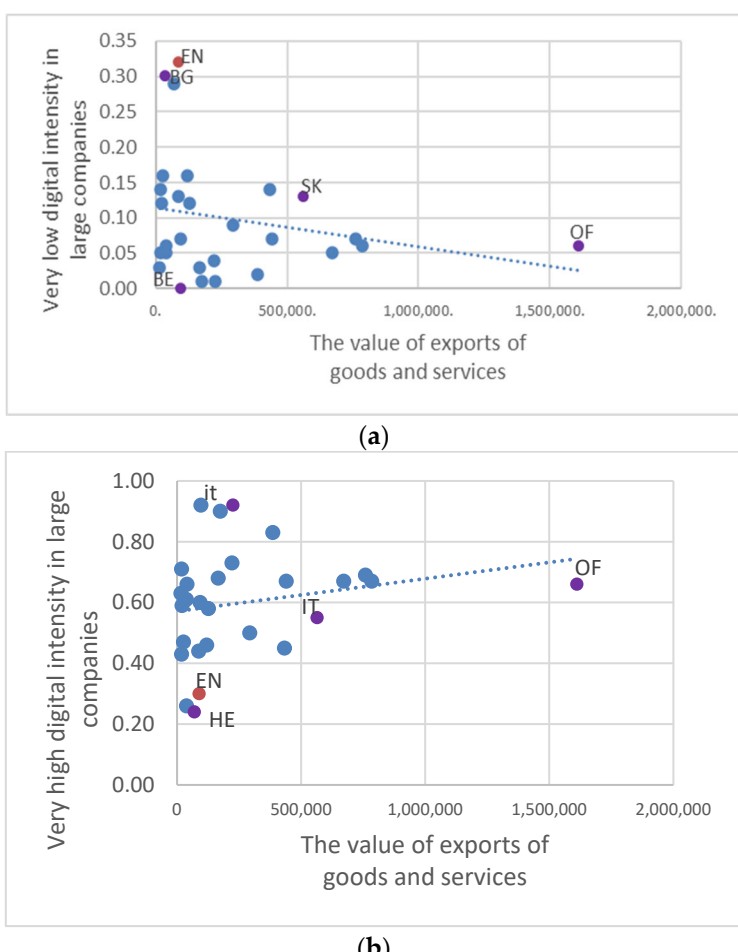

(**a**)

(**b**)

**Figure 9.** The relationship between digital intensity in large companies and exports to the EU in 2019. (**a**) The value of exports of goods and services for very low digital intensity in large companies. (**b**) The value of exports of goods and services for very high digital intensity in large companies.

## 4. Methods and Methodologies

Given that we want to know if digitalization is a predictor of labor productivity, value-added, and exports, positively influencing them, the research was formulated based on the following hypotheses:

**Hypothesis 1.** *The higher the level of digitization, the higher the labor productivity level;*

**Hypothesis 2.** *A high degree of digitalization of companies_means a positive correlation with a big value added;*

**Hypothesis 3.** *A high degree of digitalization is positively associated with a higher level of exports.*

The studied period consists of the years for which the data on digitalization are valid, more precisely the integration of digital technology, an indicator that is part of the five included in DESI (Digital Economy and Society Index), developed by the European Union: 2014–2019. The area of interest includes the states of the European Union, with an emphasis on Romania. Once the data related to all European states for the variables studied in the period 2014–2019 were obtained, they were processed, making averages of the European Union, so that Romania's situation can be compared with the European one.

For the processing of data for the validation or invalidation of variables, the SPSS program was used, with the help of which scattering, and regression diagrams were

made, and subsequently, for the validation or invalidation of the proposed econometric model based on the results obtained in this stage, the program EViews 13 (IHS Global Inc., 4521 Campus Drive, #336, Irvine, CA 92612-2621, USA) was used.

For this research, we opted for regression to the detriment of the correlation because it can determine how one variable causes the change of another, while in the case of correlation, the two variables are interchangeable, which is not the case here, given the hypotheses previously expressed.

The general form of the simple linear regression model is $Y = \beta_0 + \beta_1 X + \varepsilon$, where Y represents the independent variable, X the dependent variable, $\beta_0$ and $\beta_1$ regression coefficients (ordered at the origin, respectively the slope of the line), and $\varepsilon$ is the random variable, which indicates the impact of other variables on the dependent variable. This general form takes the following forms in the case of this research:

$$DG_{EU} = \beta_0 + \beta_1\, PM_{EU} + \varepsilon \text{ and } DG_{RO} = \text{respectively } \beta_0 + \beta_1\, PM_{RO} + \varepsilon$$

$$DG_{EU} = \beta_0 + \beta_1\, VAB_{UE} + \varepsilon,\ DG_{UE} = \beta_0 + \beta_1\, VAB_{RO} + \varepsilon$$

$$DG_{EU} = \beta_0 + \beta_1\, EX_{UE} + \varepsilon,\ DG_{RO} = \beta_0 + \beta_1\, EX_{RO} + \varepsilon$$

First, a regression was performed to find out whether digitalization can be used as a predictor for the three dependent variables. This was done both for the data related to the European Union (averaging them) and for Romania. To achieve linear regressions, a linear relationship was needed between the two variables, which can be tested using the *scatterplot*, the graph being inspected for linearity (Scheme 1a–c).

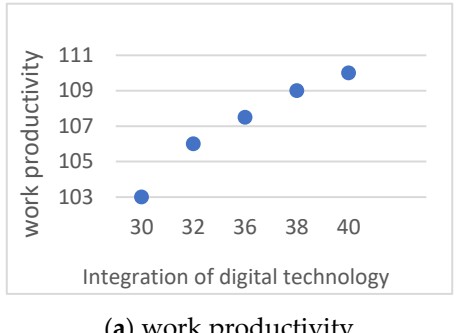 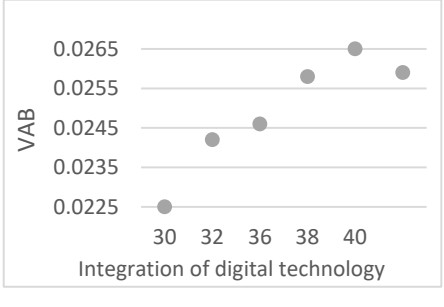 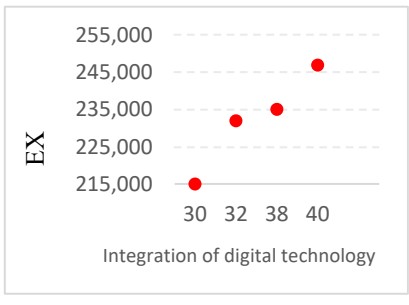

(**a**) work productivity   (**b**) VAB   (**c**) EX

**Scheme 1.** Distribution charts-media of EU countries (2014–2019) (**a**) labor productivity, (**b**) gross value added, (**c**) exports.

The representative scatterplot for the relationship between digitization and labor productivity, between 2014 and 2019 for the average of the European Union states, indicates an almost perfect distribution, indicating that these data are ideal to be analyzed by the linear regression method.

A good distribution for this purpose is both the gross value added and the exports, indicating that in all three cases, linear regression can be used. This visual representation suggests a possible positive correlation between digitalization and the three dependent variables (Scheme 2a–c).

For Romania, the scatter plots do not indicate an equally clear situation, the residues being higher than in the case of the spread plots for the European Union, but the distribution is clear enough for the operation of linear regression.

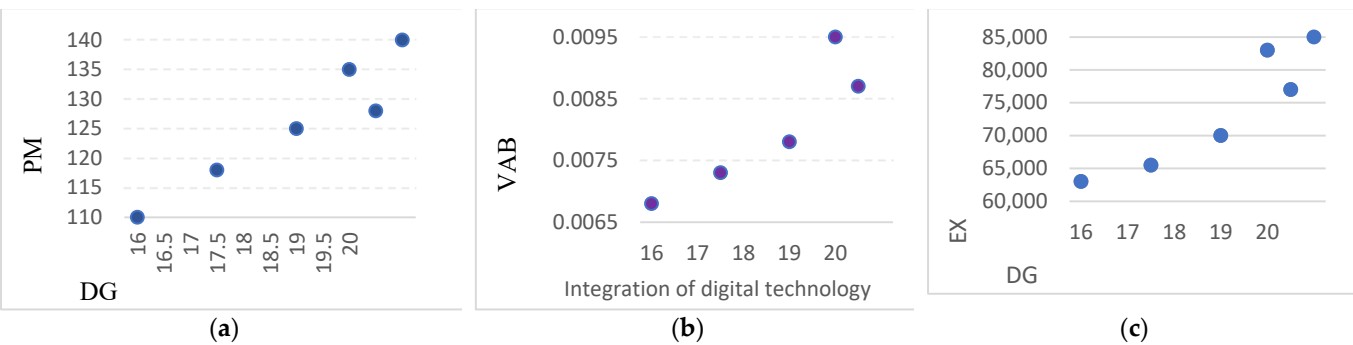

**Scheme 2.** Spread charts—Romania (2014–2019) (**a**) labor productivity, (**b**) gross value added, (**c**) exports.

## 5. The Results of the Research

### 5.1. Digitalization and Labor Productivity

Using these methods, a high correlation has been achieved between digitalization and labor productivity in the case of the European Union and, moreover, a high impact of the integration of digital technology on labor productivity. The coefficient of determination for regression $R^2$ (which is, in fact, identical to the coefficient of determination for correlation), is 0.98, which means that 98% of labor productivity in the European Union between 2014 and 2019 can be explained by integration of digital technologies—the correction of $R^2$ slightly decreases this percentage, up to 97%.

The result of the analysis of the sources of variation in terms of the dependent variable, labor productivity, and the validity of the regression model, indicates two important aspects: first, the ratio $F (190, 48)$ has a higher value than the residual influence, and second, the value of sig. it is much smaller than 0.05, being 0.000, which indicates the statistical significance of the tested model (Figure 10).

### Model Summary (Work productivity)

| R | R Square | Adjusted R Square | Std. Error of the Estimate |
|---|---|---|---|
| .99 | .98 | .97 | .53 |

### ANOVA (Work productivity)

| | Sum of Squares | df | Mean Square | F | Sig. |
|---|---|---|---|---|---|
| Regression | 53.70 | 1 | 53.70 | 190.48 | .000 |
| Residual | 1.13 | 4 | .28 | | |
| Total | 54.83 | 5 | | | |

### Coefficients (Work productivity)

| | Unstandardized Coefficients | | Standardized Coefficients | t | Sig. |
|---|---|---|---|---|---|
| | B | Std. Error | Beta (β) | | |
| (Constant) | 85.52 | 1.88 | .00 | 43.94 | .000 |
| Integration of digital technology | .71 | .05 | .99 | 13.80 | .000 |

**Figure 10.** Linear regression: digitization and labor productivity—an average of European Union countries.

In the table related to the coefficients, there is the information necessary to predict labor productivity according to digitalization, but also the degree to which digitization contributes to the model. This results in the following regression equation for predicting digitalized labor productivity in the European Union:

$$PM_{UE} = 82.52 + 0.71 \times DG_{UE}$$

When we perform the same operation for the data related to Romania (Figure 11), we obtain a statistically significant model, but also a lower impact of digitization on labor productivity.

## Model Summary (Work productivity)

| R | R Square | Adjusted R Square | Std. Error of Estimate |
|---|---|---|---|
| .93 | .87 | .84 | 4.56 |

## ANOVA (Work productivity)

| | Sum of Squares | df | Mean Square | F | Sig. |
|---|---|---|---|---|---|
| Regression | 553.70 | 1 | 553.70 | 26.62 | .007 |
| Residual | 83.23 | 4 | 20.81 | | |
| Total | 637.07 | 5 | | | |

## Coefficients (Work productivity)

| | Unstandardized Coefficients | | Standardized Coefficients | | |
|---|---|---|---|---|---|
| | B | Std. Error | Beta (β) | t | Sig. |
| (Constant) | 16.05 | 21.77 | .00 | .74 | .494 |
| Integration of digital technology | 5.90 | 1.14 | .93 | 5.16 | .007 |

**Figure 11.** Linear regression: digitalization and labor productivity—Romania.

The adjusted regression coefficient is 0.84—therefore, 84% of labor productivity in Romania can be explained by digitization, and the standard deviation is much higher than in the European Union. The significance coefficient is 0.007, which indicates the statistical significance of the test. Given the data obtained in the table of coefficients, we obtain the following regression equation:

$$PM_{RO} = 16.05 + 5.90 \times DG_{RO}$$

*5.2. Digitalization and Gross Value Added*

In the case of the effect of integrating digital technologies on gross value added, different results have been obtained. Even though the scatter plot seemed to suggest a strong relationship between the two variables in the case of the European Union (Figure 12), the significance coefficient is very high (0.672), indicating a statistically insignificant relationship between digitization and gross value added. in the European Union. In fact, the correlation coefficient is also low (0.05), and, after correcting it, it becomes a negative one (−0.19).

## Model Summary (VAB)

| R | R Square | Adjusted R Square | Std. Error of the Estimate |
|---|---|---|---|
| .22 | .05 | -.19 | 32402.58 |

## ANOVA (VAB)

| | Sum of Squares | df | Mean Square | F | Sig. |
|---|---|---|---|---|---|
| Regression | 218752806.80 | 1 | 218752806.80 | .21 | .672 |
| Residual | 4199709568.52 | 4 | 1049927392.13 | | |
| Total | 4418462375.32 | 5 | | | |

## Coefficients (VAB)

| | Unstandardized Coefficients | | Standardized Coefficients | | |
|---|---|---|---|---|---|
| | B | Std. Error | Beta (β) | t | Sig. |
| (Constant) | 401057.93 | 114589.59 | .00 | 3.50 | .017 |
| Integration of digital technology | 1433.65 | 3140.83 | .22 | .46 | .672 |

**Figure 12.** Linear regression: digitalization and gross value added—an average of EU states.

Looking at g the relationship between digitization and gross value added in Romania (Figure 13), the situation is less clear. On the one hand, $R^2$ has a value of 0.81, which would normally indicate that digitization can explain 81% of Romania's gross value added. At

the same time, the significance coefficient is one in parameters, even being close to the upper limit, having the value of 0.037. On the other hand, the regression equation cannot be calculated, the resulting formula not being valid:

$$VAB_{RO} = 0.00 + 0.00 \times VAB_{RO}$$

## Model Summary (VAB)

| R | R Square | Adjusted R Square | Std. Error of the Estimate |
|---|---|---|---|
| .90 | .81 | .75 | .00 |

## ANOVA (VAB)

| | Sum of Squares | df | Mean Square | F | Sig. |
|---|---|---|---|---|---|
| Regression | .00 | 1 | .00 | 12.81 | .037 |
| Residual | .00 | 3 | .00 | | |
| Total | .00 | 4 | | | |

## Coefficients (VAB)

| | Unstandardized Coefficients | | Standardized Coefficients | | |
|---|---|---|---|---|---|
| | B | Std. Error | Beta (β) | t | Sig. |
| (Constant) | .00 | .00 | .00 | -.72 | .512 |
| Integration of digital technology | .00 | .00 | .90 | 3.58 | .037 |

**Figure 13.** Linear regression: digitalization and gross value added—Romania.

### 5.3. Digitalization and Exports

The simple linear regression in the case of the effect of digitization and exports to the European Union obtained a coefficient of determination for regression of 0.71 and, adjusted, 0.64. Therefore, 64% of EU exports could be explained by the integration of digital technologies; the test has statistical significance, being below the threshold of 0.05, with the value of 0.035, but we emphasize its positioning towards the upper limit of the range of statistical significance (Figure 14). Given the values in the table for coefficients, we obtain this regression equation:

$$EX_{UE} = 125,907.22 + 2982.97 \times EX_{UE}$$

## Model Summary (EX)

| R | R Square | Adjusted R Square | Std. Error of the Estimate |
|---|---|---|---|
| .84 | .71 | .64 | 9779.26 |

## ANOVA (EX)

| | Sum of Squares | df | Mean Square | F | Sig. |
|---|---|---|---|---|---|
| Regression | 947034593.85 | 1 | 947034593.85 | 9.90 | .035 |
| Residual | 382535877.82 | 4 | 95633969.45 | | |
| Total | 1329570471.66 | 5 | | | |

## Coefficients (EX)

| | Unstandardized Coefficients | | Standardized Coefficients | | |
|---|---|---|---|---|---|
| | B | Std. Error | Beta (β) | t | Sig. |
| (Constant) | 125907.22 | 34583.71 | .00 | 3.64 | .015 |
| Integration of digital technology | 2982.97 | 947.92 | .84 | 3.15 | .035 |

**Figure 14.** Linear regression: digitalization and exports—an average of EU states.

In the case of Romania, there seems to be a somewhat stronger effect of digitization on the value of exports of goods and services (Figure 15). The coefficient of determination for regression $R^2$ is equal to 0.79, and corrected to 0.73, which shows that 73% of the value of

Romania's exports can be explained with the help of digitization. The value of Mr. indicates the statistical significance of the test, and the regression equation is as follows:

$$EX_{RO} = -27{,}392.37 + 5398.54 \times EX_{RO}$$

## Model Summary (EX)

| R | R Square | Adjusted R Square | Std. Error of the Estimate |
|---|---|---|---|
| .89 | .79 | .73 | 5625.29 |

## ANOVA (EX)

| | Sum of Squares | df | Mean Square | F | Sig. |
|---|---|---|---|---|---|
| Regression | 463827918.49 | 1 | 463827918.49 | 14.46 | .019 |
| Residual | 126575593.42 | 4 | 31643898.35 | | |
| Total | 590403511.91 | 5 | | | |

## Coefficients (EX)

| | Unstandardized Coefficients | | Standardized Coefficients | t | Sig. |
|---|---|---|---|---|---|
| | B | Std. Error | Beta (β) | | |
| (Constant) | -27392.37 | 26842.86 | .00 | -1.02 | .354 |
| Integration of digital technology | 5398.54 | 1410.08 | .89 | 3.83 | .019 |

**Figure 15.** Linear regression: digitization and exports—Romania.

The value of exports of goods and services is influenced by several variables, including digitalization, mainly due to e-commerce components.

For countries such as Romania, where the value of exports and digitization are low compared to other European countries, investing in developing the capacity of e-commerce companies is an opportunity to effectively increase exports, both in volume and value.

## 6. Discussions and Proposing an Econometric Model

To point out the influence of digitalization on economic growth, there are several papers dealing with the relationship between telephone services and economic development were analyzed; the role of the telephone as a contributory agent to economic development was investigated. Dynamics analysis and cross-correlation techniques showed that the telephone systematically contributed to economic development. The antiquated state of the telecommunications network in the transitional economies of Central and Eastern Europe has been identified by the OECD (1993) and the ITU (1994) as a significant impediment to regional productivity, international competitiveness, and trade performance. This situation suggests that the upgrading and extension of the telecommunications network should be a priority objective for policymakers to facilitate growth [19–21]. A direct connection was found between access to fixed broadband Internet and GDP growth per capita in both developed and developing countries and analysis was carried out for a group of 192 countries and revealed that investments in mobile telecommunications infrastructure made a considerable contribution to economic and productivity growth [22–25].

Following the panel analysis of the data related to a number of 116 countries in the period 2014–2019, the paper investigates the association between broadband internet speed and labor productivity. The authors identified a significant and robust relationship when a one-year lag is introduced for the series defining mobile broadband infrastructure. Interpretation of the results shows that a 10% increase in mobile broadband infrastructure in period $t-1$ is associated with a 0.2% increase in labor productivity in period t. The results are robust only to non-OECD countries and with low income [26–28].

Of all the variables tested, the strongest results were obtained in the case of the effect of digitalization on labor productivity, both in the case of the EU average and that of Romania. Therefore, it was decided to propose an econometric model, which should be tested using panel data (for all European Union countries and the period 2014–2019).

The specialty literature highlights several factors that have a positive impact on labor productivity, such as technical efficiency, which depends on both the tools used and

the capital used in investments, capital goods, specialization, new technologies, human capital, and more [19]. The number of possible variables with a positive impact on labor productivity is high, but at one point they overlap. From these, the following variables were selected with a positive impact demonstrated in the literature on labor productivity:

- ED: level of education and specialization of the adult population, expressed as a proportion of adults aged between 25 and 64 years old who were involved in formal or non-formal learning activities 4 weeks before collecting data (data obtained from Eurostat, under the code 'sdg_04_60');
- INV: investments measured as gross fixed capital formation (GFCF) relative to gross domestic product, expressed in percentage points;

The least squares method is a common way to perform a multiple regression analysis (Figure 16); In the case of the linear one, the objective is that the points related to the variables match a line, the best matching line being the one in which the deviations are minimal. The purpose of performing the regression analysis is to answer the question of whether the selected variables have an impact on the dependent variable, especially if it is a positive one. In EViews, the test selected was *Least Panels Squares*, with standard *cross-section weights* (PCSE) options *errors & covariance*.

| Variable | Coefficient | Std. Error | t-Statistic | Prob. |
|----------|-------------|------------|-------------|-------|
| C | 88.45051 | 3.407768 | 25.95555 | 0.0000 |
| DG | 0.276313 | 0.078252 | 3.531063 | 0.0006 |
| ED | -0.361793 | 0.332826 | -1.087032 | 0.2789 |
| INV | 0.122226 | 0.041405 | 2.951970 | 0.0037 |

| Effects Specification | | | |
|---|---|---|---|

Cross-section fixed (dummy variables)

| Root MSE | 3.127046 | R-squared | 0.889066 |
|----------|----------|-----------|----------|
| Mean dependent var | 108.0509 | Adjusted R-squared | 0.864595 |
| S.D. dependent var | 9.416864 | S.E. of regression | 3.465158 |
| Akaike info criterion | 5.489312 | Sum squared resid | 1632.996 |
| Schwarz criterion | 6.068101 | Log likelihood | -427.3576 |
| Hannan-Quinn criter. | 5.724230 | F-statistic | 36.33184 |
| Durbin-Watson stat | 0.818936 | Prob(F-statistic) | 0.000000 |

**Figure 16.** Least squares method—the effect of digitalization, education, and investment on labor productivity (EViews).

The proposed equation is as follows:

$$PM_{it} = f\ (DG_{it},\ ED_{it},\ INV_{it,C})$$

where PM represents labor productivity, DG digitization, ED education, INV investment, and C constant.

Of the three independent variables, only two were statistically significant: digitalization, with a value of 0.0006, and investments, with 0.0037. On the other hand, surprisingly, education in this form was not a variable with a measurable effect on labor productivity, having a *probative value* of 0.2789.

The coefficients of the variable's digitalization (0.276313) and investments (0.122226) have positive values, which means that they have a positive contribution to increasing labor productivity.

Another relevant indicator is $R^2$, the square of the multiple correlation coefficient. It shows the percentage by which the variation of the dependent variable can be explained by the equation. $R^2$ increases as the number of variables included in the model increases; therefore, we consider that its value for the indicated model is high, 86.45% of the labor

productivity in the EU states can be explained by this model. However, the high value of the standard error must also be considered.

Following the test, the following formula for predicting labor productivity for the Member States of the European Union was obtained:

$$PM = 88.45051 + (0.276313 \times DG) + (-0.361793 \times ED) + (0.122226 \times INV)$$

So, in this research, the effect of integrating digital technologies on labor productivity, gross value added, and exports were tested. The most robust results were obtained in the case of labor productivity, which led us to propose an econometric model that considers the integration of other elements indicated in the literature: adult education and training, but also the level of investment. The proposed model confirmed the impact of digitalization and investment on productivity, but not on the indicator of lifelong learning.

## 7. Conclusions

Digitalization, robotics, automation, process integration, and many other technological advances that may seem, at first sight of a technological nature, depend on the human factor—on how well adapted the human resource is to the profound changes of the new Industrial Era. For Romania to have a competitive workforce in the context of *Industry 4.0*, many issues need to be improved.

The main conclusion regarding the relationship between digitalization and labor productivity is that the benefits of digitalization have not spread evenly among companies, with the advantage of those with superior access to technical, managerial, and organizational skills. In fact, it is about those companies that already tended to be more productive than the average, and digitalization has helped to increase their advantage. Also, the impact of digitalization on labor productivity is different depending on the industry to which the company belongs, the intensively digitalized sectors being advantageous. At the same time, digitalization is not really a homogeneous trend, but includes a diverse set of technologies, with different impacts on the economy in general and the labor market. Productivity is also subject to many influences, which must be considered in such studies. In conclusion, digitalization contributes to increasing the productivity of companies, when they meet a few conditions, but it does not necessarily represent a discriminatory effect.

Using data provided by Eurostat and the European Commission, the impact of digitization in the Member States of the European Union was analyzed between 2014 and 2019. This correlation also exists between digitization and exports, although it is weaker than in the first cases. This result can be explained by the small number of companies investing in international e-commerce, considering that as their number increases, the impact of digitization on the total value of exports could be much easier to notice.

In Romania, statistics show, on the one hand, a low level of digitalization and, on the other hand, low values of labor productivity, gross value added per capita, and exports. Given the positive impact of digitalization on them, Romania's economic situation is defining in the next period.

Increasing the digitalization of companies is not enough, given that digitalization indicators are constantly improving in the EU. Maintaining the slow pace of growth of Romania's digitalization would even lead to deepening digital and economic differences between Romania and the rest of the European Union.

Instead, strong and targeted growth is a special opportunity for the Romanian economy, as follows: increasing the number of digital technologies used by SMEs and large companies, along with employee skills, to improve labor productivity, but also increase the number of companies using trade for international trade, to increase the total value of exports.

Given the importance that digitalization already has for the whole economy as well as the labor market, it can be considered that further research and analysis are needed. The clear illustration of the impact of digitalization extends beyond the important theoretical but also practical implications, influencing both the decision-making process of

company management (whether small, medium, or large), as well as national and European economic policies.

**Author Contributions:** Conceptualization, A.Ș.C. and A.I.C.; methodology, A.Ș.C. and A.I.C.; validation, A.B., D.V. and T.N.G.; formal analysis, R.C.D.; resources, M.R.S.-P. and A.B.; writing—original draft preparation, M.R.S.-P., N.M. and L.T.; writing—review and editing, A.Ș.C., N.M. and A.B.; visualization, A.Ș.C., D.V., A.M.M., A.B., N.M. and T.N.G. All authors have read and agreed to the published version of the manuscript.

**Funding:** This research received no external funding.

**Data Availability Statement:** Not applicable.

**Conflicts of Interest:** The authors declare no conflict of interest.

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
