# Peer review of "The Impact of Digitalization on Macroeconomic Indicators in the New Industrial Age"

_electronics, doi:10.3390/electronics12071612_

Round 1

Reviewer 1 Report

1. The sentence is incomplete in lines 62-63. 

2. Resize the charts in lines 184-187 to avoid situation in 188 , where the caption in Figure      9 appears alone at the top of a new page. Similar situations should be avoided.

3. What is the basis for choosing digital impact indicators? It is suggested to increase the         application and comparison of mature economic models to new technologies in 

     published literature.

Author Response

  1. The sentence is incomplete in lines 62-63. 

R: We made the correction.

  1. Resize the charts in lines 184-187 to avoid situation in 188, where the caption in Figure      9 appears alone at the top of a new page. Similar situations should be avoided.

R: We made the correction.

  1. What is the basis for choosing digital impact indicators? It is suggested to increase the application and comparison of mature economic models to new technologies in published literature.

R: We add more information regarding topic that reviewer asked. Please find tha revised version of our paper.

Thank you very much for your valuable comments!

Reviewer 2 Report

The aim of this interesting study was to show the impact of digitalization on macroeconomic indicators, including labor productivity, value-added, and value of exports of goods and services, and in this regard, we have built an econometric model. The added value of the study is the detailed statistical correlations between the integration of digital technologies and labor productivity, gross value added, and exports - at the EU level.

The indicator chosen to measure productivity is the productivity of labor per employee and per working hour, calculated as the actual output per unit of labor, based on the total number of working hours. The study concludes by analyzing the relationship between the degree of digitization of businesses in the Member States of the European Union and the value of exports of goods and services, experienced in millions of euros.

Three hypotheses were made in the study, which the authors confirmed with the results. The measurements and tools used by the authors appear to be valid. The results are detailed with statistical confirmation of the results.

The discussion is of appropriate length and includes the essential findings of the study. The discussion can be supplemented with additional literature to enrich the authors' arguments.

I rate the paper positively because the themes explored are highly topical and necessary. Digitalization, robotics, automation, process integration, and many other technological advances that may seem, at first sight of a technological nature, depend on the human factor – on how well adapted the human resource is to the profound changes of the new Industrial Era.

Author Response

  1. The aim of this interesting study was to show the impact of digitalization on macroeconomic indicators, including labor productivity, value-added, and value of exports of goods and services, and in this regard, we have built an econometric model. The added value of the study is the detailed statistical correlations between the integration of digital technologies and labor productivity, gross value added, and exports - at the EU level.
  2. The indicator chosen to measure productivity is the productivity of labor per employee and per working hour, calculated as the actual output per unit of labor, based on the total number of working hours. The study concludes by analyzing the relationship between the degree of digitization of businesses in the Member States of the European Union and the value of exports of goods and services, experienced in millions of euros.

  1. Three hypotheses were made in the study, which the authors confirmed with the results. The measurements and tools used by the authors appear to be valid. The results are detailed with statistical confirmation of the results.
  2. The discussion is of appropriate length and includes the essential findings of the study. The discussion can be supplemented with additional literature to enrich the authors' arguments.

R: We thank you for your appreciation and we communicate that we have implemented your valuable suggestions in the article.

Thank you very much for your valuable comments!